# Examining inequalities in access to delivery by caesarean section in Nigeria

**Boniface Ayanbekongshie Ushie[1], Ekerette Emmanuel Udoh[2], Anthony Idowu Ajayi[1] \***

**1** Population Dynamics and Reproductive Health Unit, African Population and Health Research Centre, APHRC Campus, Nairobi, Kenya, **2** Society for Family Health, Abuja, Nigeria

\* ajayianthony@gmail.com

**Data Availability Statement:** All dataset files are available from the DHS database(https://dhsprogram.com/data/available-datasets.cfm).

**Funding:** No funding was received towards completing this work.

## Abstract

### Background

Maternal deaths are far too common in Nigeria, and this is in part due to lack of access to lifesaving emergency obstetric care, especially among women in the poorest strata in Nigeria. Data on the extent of inequality in access to such lifesaving intervention could convince policymakers in developing an appropriate intervention. This study examines inequality in access to births by caesarean section in Nigeria.

### Methods

Data for 20,468 women who gave birth in the five years preceding 2013 Nigerian Demographic and Health Survey (DHS) were used for this study. Inequality in caesarean delivery was assessed using the concentration curve and multiple logistic regression models.

### Results

There was a high concentration in the utilisation of caesarean section among the women in the relatively high wealth quintile. Overall, delivery by caesarean section was 2.1%, but the rate was highest among women who had higher education and belonged to the richest wealth quintile (13.6%) and lowest among women without formal education and who belonged to the poorest wealth quintile (0.4%). Belonging to the poorest wealth quintile and having no formal education were associated with lower odds of having delivery by caesarean section.

### Conclusion

In conclusion, women in the richest households are within the WHO's recommended level of 10–15% for caesarean birth utilisation, but women in the poorest households are so far away from the recommended rate. Equity in healthcare is still a promise, its realisation will entail making care available to those in need not only those who can afford it.

**Competing interests:** The authors have declared that no competing interests exist.

## Introduction

Maternal mortality has remained high, especially in low- and middle-income countries, in spite of substantial progress recorded in the fight to improve maternal health outcomes [1]. It is estimated that about a quarter of a million women died due to pregnancy and childbirth-related complications in 2013 alone [1]. What is more, the risk of adverse maternal outcomes is not uniformly distributed because the largest proportions are concentrated in the most vulnerable and deprived groups [2]. Globally, the possibilities of surviving and living a healthy life are closely related to the socioeconomic background of individuals and families, neighbourhoods and communities, and these possibilities are reflected in a substantial and even increasing inequalities in health within countries [3]. Thus, a generalised picture of the reduction or otherwise of maternal morbidity and mortality will be misleading because of the huge disparities that exist between the poorest and the richest households in access to health care, especially for women.

Across national boundaries, the effects of inequitable access to health are quite staggering as the magnitude of morbidities and deaths emanating from resource-poor countries are overwhelmingly higher than those in resource-rich nations [4]. The disparity in access to health is even more conspicuous when analysed within each country, especially in the resource-constrained countries. The steepness of this disparity calls to question and challenges the notion of equitable access to health care that national governments uniformly claim to be implementing. Generally, there is sufficient evidence suggesting that there is a link between inequitable access to health care and inequitable distribution of illness [5]. Closing the wide gap between illness distribution and access to health for various categories of citizens, especially the most vulnerable, would increase health equity and improve the overall health of a nation. Research evidence already suggests that access to, quality and outcomes from, health care are inequitable. In fact, it has been shown that the categories of people with the greatest health challenges actually receive the lowest level of services [5]. So is equitable access to health care only a myth?

Most maternal deaths are avoidable with the use of quality obstetric services; however, many women continue to die due to lack of access to life-saving services. Increasing skilled attendance at birth, backed-up by access to antenatal care and referral-level facilities, was identified as the critical strategy to achieve the required reduction in maternal mortality [6]. Women in underserved communities lack access to crucial maternal health care services [7, 8]. For example, only 36% of births are attended by skilled health workers in Nigeria [9]. Moreover, Nigeria accounts for over 19% of global maternal deaths with an estimated 58,000 maternal deaths every year [1].

In this study, access to delivery by caesarean section (CS) is examined as the primary indicator to demonstrate inequitable access to maternal health services in Nigeria by drawing on nationally-representative data from 2013 Demographic and Health Survey (DHS). Caesarean section, as an emergency obstetric procedure, can save the lives of many women and children if used in appropriate conditions [10]; it saves women who otherwise would have to endure prolonged labour and the risk of vagina fistula and offers a lifesaving opportunity for many women during childbirth. Interestingly, reports from around the world suggest that there are significant increases in the proportion of women receiving this obstetric care [11–14]. However, it is unclear whether this is true for women in the poorest strata, especially in Nigeria. A caesarean section utilisation of 10–15% at the population level is recommended as beneficial to the health of women as it reduces neonatal and maternal mortality [10, 15]. Limited access to caesarean section is linked with an increase in maternal mortality and neonatal mortality [16, 17]. Significantly, a study has shown that the caesarean section rate of less than 1% among women in the lowest wealth index contributes about 80,000 of maternal deaths per year in sub-Saharan Africa [18]. While women in the richest households are overusing caesarean

section because they find it more convenient than to endure the pain of vaginal delivery and can afford the costs, women in the poorest households are underutilising CS because they are too poor to pay [19].

The Sustainable Development Goals (SDGs) aim to make health care universally accessible to all. Thus, there is no universal access to health care if the status quo remains, whereby women in the rich population groups over-utilise healthcare while women in the poorest strata lack access to the same care. The Nigeria government, recognising the disparity in health care utilisation, have implemented several maternal and child health care interventions aimed at making health-care universally accessible for all pregnant women and children under the age of five as a late push to achieve the Millennium Development Goals [20]. For example, the Midwives Service Scheme, which engaged newly graduated, unemployed, and retired midwives to work temporarily in rural areas, was introduced throughout the country in 2009 to bolster access to maternal health care services [21]. Also, user fee exemption for maternal health care services was introduced as part of the federal government subsidy reinvestment programme in 2011. Besides these pro-grammes, each state government introduced maternal health policies aimed at increasing access to care, especially for the poor [20]. Given this rapid scale-up of maternal health interventions in Nigeria, it is expected that poor women will have equitable access to care, including caesarean sec-tion delivery. However, the level of utilisation of birth by caesarean section among women in the poorest strata is unclear. As such, it is unknown whether births by caesarean section are a function of social status. This study, therefore, assesses the level of access to birth by caesarean section across the main socioeconomic status indicators such as education and wealth status.

## Methods

### Study population and data source

Data for this study are derived from the 2013 Nigeria Demographic and Health Survey (NDHS). The survey is a nationally representative cross-sectional household survey conducted between February and June 2013, in which a range of detailed demographic and health-related information were collected. The dataset used in this study was the women individual recode, which contains information from the women's questionnaire in addition to household infor-mation. Data for 20,468 women who had children in the five years preceding the survey were extracted and analysed in this study.

### Dependent variable

The dependent variable of this study is delivery by caesarean section among women of repro-ductive age who gave birth the five years preceding the survey. Information on caesarean deliv-ery was elicited by asking each woman if their index child was delivered by caesarean sections. Women's' responses were either "yes" or"no".

### Independent variables

The main independent variables were wealth index and education level. Wealth index was grouped into five categories from poorest to richest. Wealth index in the NDHS is computed using household characteristics, household assets, and possession of durable goods as well as access to clean water and improved sanitation. Using Principal component analysis (PCA), the wealth variable index was computed and the resulting composite score used to create five cate-gories. Level of education was measured by asking women the highest level of education they completed. The responses were categorised as 'no formal', 'primary', 'secondary' and 'higher education'.

## Covariates

Two main categories of covariates were included in the statistical modelling. The first is the demographic covariates, which include age, marital status, literacy level, membership of health insurance scheme, presence of co-wife, husband's level of education, and relationship to the household head. The demographic controls are important, given the fact that they are key determinants of health outcomes [22]. Age was grouped into six categories with mostly five years' interval, except for the age group 40–49 year. Marital status was categorised as never married, currently married and previously married. Membership of health insurance scheme was a categorical variable with a 'yes' or 'no' response.

The second category is the geographical covariates. Place of residence is one of the geographical covariates and is an important determinant of health care utilisation [20]. Place of residence was categorised into rural and urban. Health outcomes are skewed according to the geopolitical zones in Nigeria [9]. As such, the geopolitical zones were added as a covariate. Women are categorised according to the geopolitical zones where they reside.

## Data analysis

In this study, the concentration curve was used to assess socio-economic inequality in caesarean delivery in Nigeria. Absolute concentration index of inequality for caesarean delivery was also estimated. The health concentration index, first developed by Nanak Kakwani in 1977 [23], which is directly related to the concentration curve, is used to quantify the degree of socioeconomic-related inequality in a health variable. In our analysis, the formula for estimation of concentration index for grouped data was used and is as follows;

$$C = (p_1 L_2 - p_2 L_1) + (p_2 L_3 - p_3 L_2) + \ldots + (p_{T-1} L_T - p_T L_{T-1})$$

In the formula, $p_t$ is the cumulative percentage of the sample ranked by economic status in group $t$, and $Lt$ is the corresponding concentration curve ordinate. Theoretically, the maximum value of the concentration index varies between -1 and +1, but it is not limited to the range of $[-1, 1]$ if the health variable of interest takes negative or positive values. Therefore, the health outcome should be such that it is restricted to positive values [23]. The concentration index of 0 indicates no inequality.

Also, the outcome variable was cross-tabulated according to women's wealth status by maternal background characteristics. Multiple logistic regression models were estimated to understand the association between delivery by caesarean section, wealth index, and education. The first model is the baseline model, which examined the independent net effect of wealth status and education on delivery by CS. The second model included demographic covariates as control variables. In the third model, geographical covariates were further added as control variables. Interpretations of findings were based on odds ratios (OR), with OR >1 indicating a higher risk, OR < 1 indicating lower risk and OR = 1 showing no risk difference. The alpha value of 0.05 was considered to be statistically significant, and confidence interval (CI) of 95% was estimated. Statistical Package for Social Sciences (SPSS version 24.0) was used to perform the analysis. Sampling weight was applied for all univariate and bivariate analyses to account for the complex sampling adopted in the study.

## Ethical consideration

The Nigeria Health Research Ethics Committee approved the initial survey (NHREC), and all procedures were approved by the ICF Macro (Calverton, Maryland), and individual informed consent was obtained from respondents at the start of the interviews. Written permission to

use the data was obtained from Measure DHS, and since these are anonymous public data with no identifiable information of respondents, additional ethical approval was not required. The dataset is publicly available on the DHS website (https://dhsprogram.com/data/available-datasets.cfm).

## Results

### Descriptive findings

Table 1 presents a summary of the socio-demographic characteristics of the respondents. A majority of the women were Muslims (60.8%), married (94.8%), and did not have health insurance cover (98.4%).

The concentration index for caesarean delivery was a high positive value (C = 0.4855) (Table 2) with the concentration curve lying below the line of equality indicating a high concentration in the utilisation of caesarean section among the women in the relatively high wealth quintile (i.e. women in richest, richer, and middle wealth quintile) (Fig 1). If everyone, irrespective of their wealth quintiles, had the same levels of CS utilisation, the concentration curves would have been aligned with the line of equality (45-degree line).

The prevalence of caesarean delivery was 2.1% but was higher among women in the richest wealth category (7.6%) compared to women in the middle (1.5%), poor (0.9%) and poorest (0.5%) wealth categories. Among women in the richest wealth category, delivery by caesarean section was much lower among those who had lower level of education than those who had a higher educational qualification (Table 3). Having health insurance cover doubled the proportion for caesarean delivery across wealth status except for those in the poorer wealth category.

### Multivariate findings

Three models were fitted to examine inequality in access to birth by caesarean section. The first model is the baseline model, which is used to examine the net effect of wealth index and educational status on having a birth by caesarean section. The results of Model 1 indicate that wealth index and education status were positively and significantly associated with having a birth by caesarean section. In the second model, demographic controls were included to observe how the effect of education and wealth on delivery by caesarean section changes. The results show that the magnitude and direction of the effect persist despite adding the demographic controls. The third model included geographical covariates. The magnitude of the effect of education and wealth on birth by caesarean section slightly reduces, but the direction of the effect remains the same. Delivery by caesarean section was significantly less likely to occur among women in the poorest, poor, middle wealth categories compared with those in the wealthiest category, as shown in the three models in Table 4. Women who had no formal education were less likely to undergo caesarean section compared to women who had higher education.

## Discussion

This study examined inequality in access to emergency obstetric care, specifically delivery by caesarean section in Nigeria. The findings show a significant level of inequality in access to child delivery by caesarean section with a high concentration of use among women of higher socio-economic status. High absolute inequality in birth by caesarean section has been observed in other countries in Africa (Boatin *et al.*, 2018). The study found that women in the poorest households had less than 1% level of use of CS births. Caesarean section rate of less than 1% among women in the lowest wealth index has been said to contribute to about 80,000

**Table 1. Weighted distribution of women by socio-demographic, maternal and obstetric outcome.**

| Characteristics | n | % |
|---|---|---|
| **Age** | | |
| 15–19 | 1323 | 6.5 |
| 20–24 | 4009 | 19.6 |
| 25–29 | 5376 | 26.3 |
| 30–34 | 4247 | 20.7 |
| 35–39 | 3173 | 15.5 |
| 40–49 | 2340 | 11.4 |
| **Marital status** | | |
| Never married | 453 | 2.2 |
| Currently married | 19397 | 94.8 |
| Previously married | 617 | 3.0 |
| **Religion** | | |
| Christian | 7713 | 37.7 |
| Muslim | 12436 | 60.8 |
| others | 318 | 1.6 |
| **Presence of co-wife** | | |
| No | 13067 | 64.0 |
| Yes | 7355 | 36.0 |
| **Relationship to household head** | | |
| Head | 1373 | 6.7 |
| Wife | 17052 | 83.3 |
| Others | 2042 | 10.0 |
| **Education** | | |
| No formal | 9794 | 47.9 |
| Primary | 3915 | 19.1 |
| Secondary | 5475 | 26.7 |
| Higher | 1283 | 6.3 |
| **Husbands education** | | |
| No formal | 7785 | 39.3 |
| Primary | 3661 | 18.5 |
| Secondary | 5806 | 29.3 |
| Higher | 2566 | 12.9 |
| **Wealth quintile** | | |
| Poorest | 4699 | 23.0 |
| Poor | 4588 | 22.4 |
| Middle | 3902 | 19.1 |
| Rich | 3674 | 18.0 |
| Richest | 3604 | 17.6 |
| **Health insurance coverage** | | |
| No | 20063 | 98.4 |
| Yes | 321 | 1.6 |
| **Place of residence** | | |
| Urban | 7278 | 35.6 |
| Rural | 13189 | 64.4 |
| **Parity** | | |
| 1 | 3670 | 17.9 |
| 2 | 3361 | 16.4 |

(*Continued*)

**Table 1.** (Continued)

| Characteristics | n | % |
|---|---|---|
| 3 | 3054 | 14.9 |
| 4 | 2732 | 13.3 |
| 5+ | 7650 | 37.4 |
| Literacy | | |
| No | 11817 | 58.3 |
| Yes | 8465 | 41.7 |

maternal deaths per year in sub-Saharan Africa [18]. The pattern revealed in the current study indicates low access to birth by caesarean section. Poor access to delivery by caesarean section remains mainly an issue among the economically and the socially deprived, which has serious implications for reducing maternal mortality and morbidity in the country. Nigeria remains a largely unequal society with the majority of the population living in poverty [24]. Access to health care, especially birth by caesarean section, remains elusive for many women in the poorest households.

The drivers of inequality in access to birth by caesarean section in Nigeria are two-fold, demand and supply related factors. Illiteracy, lack of formal education and poverty are demand-related factors that influence access to lifesaving interventions such as caesarean section, thus, resulting in an increased level of maternal mortality. Women in the richest quintile who are educated up to post-secondary school level had a caesarean section rate above 13%, which is within the WHO's recommended level of ideal rate of birth by caesarean section [10]. However, women in the poorest quintile who had no formal education had a caesarean section rate of 0.4%. Even among women in the richest wealth category who are uneducated, the level of birth by caesarean section is below 10%. Being poor does not immune women from experiencing birth complications, rather the intersectionality of poverty and poor access to antenatal care may increase the risks of complications requiring an emergency caesarean section. Having the resources to access emergency obstetric interventions, such as CS promptly, could mean that the lives of women and neonates are saved.

Research shows that the cost of CS is one of the main reasons women refuse birth by CS against competent medical advice [25]. Women in the poorest stratum are unable to afford the cost of CS, and instead, as soon as these women become pregnant, they and their family members begin a series of fasting and prayers for God to intervene and ensure they experience vaginal delivery even if it means they have to endure prolonged labour. Some studies have established that ability to pay for CS is one of the factors associated with having a caesarean birth [26–28] which suggests that providers may see caesarean sections as a money-making

**Table 2. Caesarean delivery and concentration index for inequality among women in Nigeria.**

| Wealth group | No. of women 15–49 years | Relative % Population women 15–49 years | Cumulative % Population women 15–49 years | No. of CS | Relative % CS | Cumulative % CS | Conc. Index (C) |
|---|---|---|---|---|---|---|---|
| | | | 0% | | | 0% | |
| Poorest | 4699 | 23% | 23% | 25 | 5% | 5% | 0.0088 |
| Poor | 4588 | 22% | 45% | 43 | 9% | 14% | 0.0293 |
| Middle | 3902 | 19% | 64% | 60 | 12% | 26% | 0.0721 |
| Rich | 3674 | 18% | 82% | 90 | 19% | 45% | 0.3754 |
| Richest | 3604 | 18% | 100% | 268 | 55% | 100% | 0.0000 |
| **Total** | **20467** | | | **486** | | | **0.4855** |

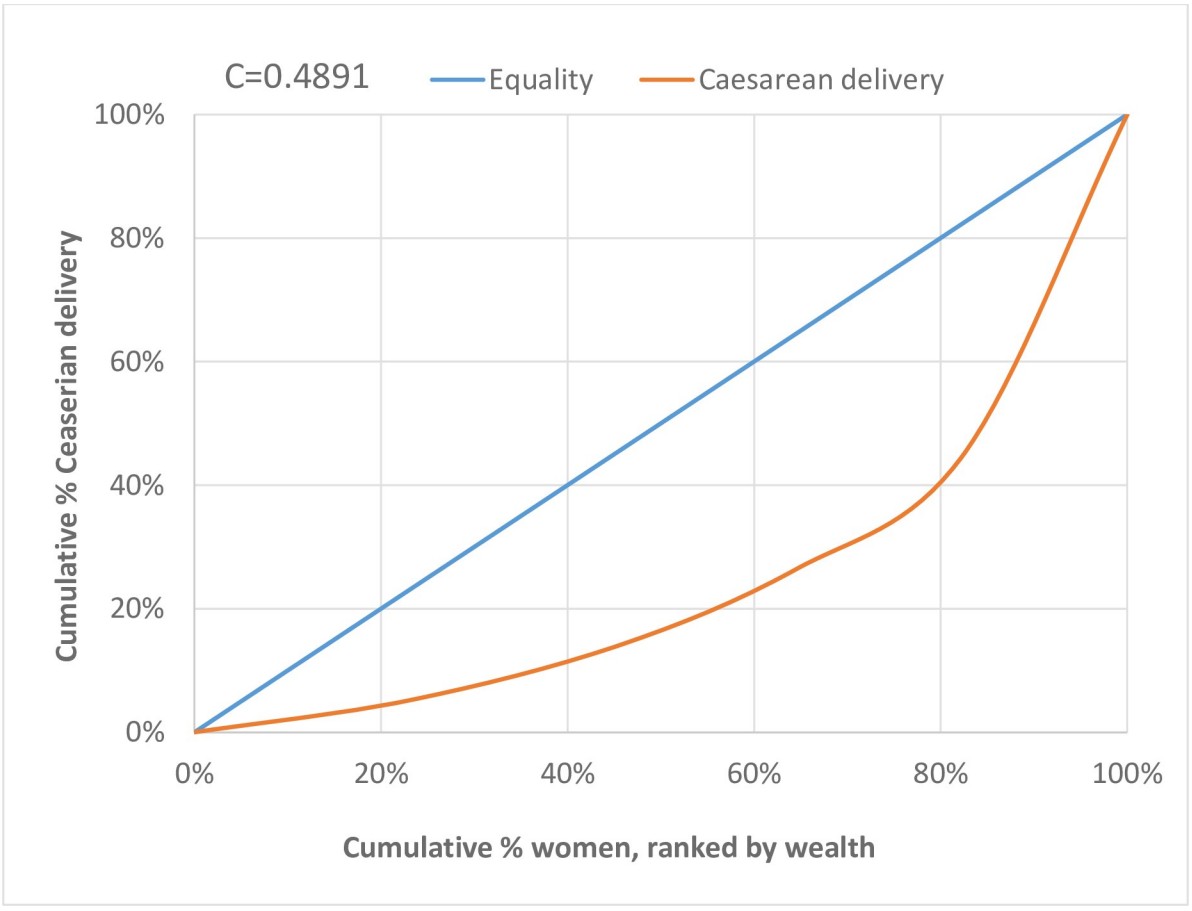

**Fig 1. Concentration curve for cumulative caesarean delivery by wealth quintile in Nigeria.**

mechanism and mainly targeted at clients who can afford the service. As such, some of the elective caesarean sections are not necessary. Moreover, proof of payment or ability to pay is required for the procedure to commence, creating a problem for women in the low-income category. In some circumstances, the emergency CS is provided, but women are detained in the hospital until they have cleared the bill [29]. Caesarean section also increases the length of hospital stay after delivery, considering the longer time for recovery from the surgery compared to vaginal delivery [30]. Longer stays in the hospital increases the overall cost of delivery–issues that women grapple with when deciding on accepting a recommendation to undergo an emergency CS [30].

The cost of birth by caesarean section in Nigeria and the lack of social and financial protection means that this life-saving intervention is unavailable to poor women. The National Health Insurance Scheme was established since 2005 to provide social and financial risk protection by reducing the cost of health care and providing equitable access to basic health services [31]. However, health insurance coverage in Nigeria is less than two percent, with most subscribers being educated and working-class women, and almost all women in the lowest wealth cluster having no insurance covers. Expanding access to the national health insurance could be one of the strategies to address inequity in delivery by caesarean section in the country.

**Table 3. Distribution of Caesarean section by wealth status and selected characteristics.**

| Characteristics | Caesarean section | | | | |
|---|---|---|---|---|---|
| | Poorest wealth group | Poor wealth group | Middle wealth group | Rich wealth group | Richest wealth group |
| | % | % | % | % | % |
| **Age** | | | | | |
| 15–19 | 1.1 | 1.7 | 0.8 | 1.3 | 5.4 |
| 20–24 | 0.2 | 1.3 | 2.0 | 2.0 | 4.2 |
| 25–29 | 0.6 | 0.6 | 1.4 | 2.4 | 6.8 |
| 30–34 | 0.5 | 0.6 | 1.3 | 4.0 | 6.7 |
| 35–39 | 0.4 | 0.6 | 1.4 | 1.9 | 12.2 |
| 40–49 | 0.5 | 1.1 | 2.1 | 2.1 | 8.6 |
| **Marital status** | | | | | |
| Never married | 3.8 | 3.7 | 2.0 | 3.6 | 5.1 |
| Currently married | 0.5 | 0.9 | 1.5 | 2.4 | 7.5 |
| Previously married | 0.0 | 2.0 | 1.5 | 1.7 | 12.5 |
| **Religion** | | | | | |
| Christian | 1.8 | 2.1 | 2.2 | 3.5 | 9.2 |
| Muslim | 0.4 | 0.5 | 0.9 | 1.4 | 4.3 |
| Others | 0.0 | 0.0 | 5.6 | 0.0 | 7.4 |
| **Presence of co-wife** | | | | | |
| No | 0.5 | 1.0 | 1.6 | 2.6 | 8.0 |
| Yes | 0.5 | 0.9 | 1.4 | 2.2 | 5.9 |
| **Relationship to household head** | | | | | |
| Head | 0.0 | 1.6 | 0.6 | 3.4 | 8.4 |
| Wife | 0.4 | 0.7 | 1.4 | 2.3 | 7.4 |
| Others | 1.8 | 2.8 | 2.7 | 2.9 | 8.4 |
| **Education** | | | | | |
| No formal | 0.4 | 0.6 | 0.6 | 0.8 | 2.4 |
| Primary | 1.2 | 1.3 | 2.2 | 2.4 | 2.8 |
| Secondary | 2.0 | 2.4 | 2.2 | 2.8 | 6.1 |
| Higher | 0.0 | 12.5 | 1.6 | 6.1 | 13.6 |
| **Husbands education** | | | | | |
| No formal | 0.4 | 0.6 | 0.5 | 0.7 | 1.1 |
| Primary | 0.3 | 1.2 | 1.2 | 2.2 | 3.0 |
| Secondary | 2.1 | 1.4 | 2.2 | 2.4 | 6.5 |
| Higher | 2.9 | 1.3 | 2.7 | 4.0 | 10.9 |
| **Parity** | | | | | |
| 1 | 1.1 | 2.7 | 3.1 | 3.4 | 10.6 |
| 2 | 0.3 | 0.4 | 1.4 | 3.0 | 6.7 |
| 3 | 0.3 | 0.4 | 1.4 | 3.0 | 6.7 |
| 4 | 0.5 | 0.9 | 0.6 | 3.1 | 6.4 |
| 5+ | 0.4 | 0.5 | 0.8 | 1.7 | 6.0 |
| **Health insurance coverage** | | | | | |
| No | 0.5 | 0.9 | 1.5 | 2.4 | 7.1 |
| Yes | 0.0 | 0.0 | 4.3 | 6.8 | 14.3 |
| **Place of residence** | | | | | |
| Urban | 1.7 | 1.8 | 1.7 | 2.6 | 7.8 |
| Rural | 0.5 | 0.8 | 1.5 | 2.4 | 5.9 |
| **Literacy** | | | | | |

(*Continued*)

**Table 3.** (Continued)

| Characteristics | Caesarean section | | | | |
| --- | --- | --- | --- | --- | --- |
| | Poorest wealth group | Poor wealth group | Middle wealth group | Rich wealth group | Richest wealth group |
| | % | % | % | % | % |
| No | 0.5 | 0.8 | 1.1 | 1.3 | 1.3 |
| Yes | 1.7 | 1.5 | 2.2 | 3.0 | 8.2 |
| **Total** | **0.5** | **0.9** | **1.5** | **2.5** | **7.6** |

Also, having a higher education could mean that women recognise the importance of having a caesarean birth when medically indicated. There is evidence that some women refuse or delay the decision to have a delivery by caesarean section [32] in the hope that they will eventually have a vaginal birth, which contributes to poor outcomes of caesarean deliveries in sub-Saharan Africa [33]. Health literacy is generally higher among women who have higher education compared to those without formal education [34–36]. Studies have established that women, especially the poor and uneducated, are afraid of CS, thus refusing it and rather seeking spiritual intervention to avoid it in Nigeria [37, 38]. Women take pride in natural delivery and attribute birth complications requiring CS to "evil forces", which must be combated by prayers [39]. Our study shows that the rate of caesarean birth was low among women in the highest wealth index who are uneducated, further highlighting the importance of education on maternal health outcomes. Other reasons why women may not accept CS include the fear of resulting pains, scars and the risk of infections. In Nigeria, it is commonplace for mothers and mother-laws to spend time with their daughters after delivery to perform hotpresses on the stomach to ensure the woman's stomach does not become distended as a result of pregnancy and childbirth. Given the intersection of wealth and education and its importance to maternal health outcomes, there is a need for a robust national strategy on women empowerment.

The reasons for inequality in delivery by caesarean section are, however, not limited to demand-related factors but are also supply-related. Use of emergency caesarean section depends on the availability of services that are of sufficient quality. In many rural settings in Nigeria, there are no health facilities [40], and in areas where they exist, the quality of services available is poor, with a lack of consistent supplies, equipment and human resources, especially doctors [21, 41–45]. The problem is further exacerbated by insufficient referral systems and lack of a functioning ambulatory system or a quick alternate transport system [46, 47]. The lack of a well-coordinated referral system with adequate means of transportation causes a delay in accessing emergency obstetric care, especially for women in underserved communities resulting in adverse outcomes not only for mothers but also the babies.

The poor quality of maternal health care in Nigeria is well documented [41, 48]. Poor quality of care results in adverse outcomes, erodes the confidence of users and may also explain why the use of caesarean section is low in the country. A recent Lancet commission study reveals that over 50% of maternal deaths in sub-Saharan Africa occur due to poor quality of care [49, 50]. Also, a recent study has shown that outcomes of caesarean section are poor in sub-Africa, with maternal deaths following C-section 50 times higher in Africa compared to high-income countries [51]. Women in Nigeria are aware of the adverse outcomes associated with caesarean section and may explain why women are afraid of caesarean section [37, 38]. Training and equipping doctors in Nigeria to provide quality obstetric services is required to restore users confidence and save the lives of mothers and babies.

Given the findings of this study, it is critical to address the inequality in the use of caesarean section in Nigeria and by extension, inequality in access to maternal health care services.

**Table 4. Logistic models for delivery by caesarean section.**

| Factors | Delivery by caesarean section | | |
|---|---|---|---|
| | Model 1 | Model 2 | Model 3 |
| **Wealth index** | | | |
| Poorest | 0.25 (0.14, 0.39)*** | 0.41 (0.24, 0.97)** | 0.58 (0.28, 0.92)* |
| Poor | 0.33 (0.21, 0.45)*** | 0.44 (0.34, 0.92)*** | 0.6 (0.38, 0.95)* |
| Middle | 0.40 (0.26, 0.48)*** | 0.46 (0.35, 0.78)*** | 0.58 (0.38, 0.91)** |
| Rich | 0.44 (0.34, 0.57)*** | 0.52 (0.38, 0.75)*** | 0.58 (0.39, 0.80)*** |
| Richest | 1 | 1 | 1 |
| **Education** | | | |
| No formal | 0.11 (0.09, 0.20)*** | 0.42 (0.26, 0.95)* | 0.48 (0.29, 1.48)* |
| Primary | 0.29 (0.24, 0.46)*** | 0.66 (0.43, 1.28) | 0.67 (0.44, 1.31) |
| Secondary | 0.44 (0.39, 0.62)*** | 0.68 (0.57, 0.75)** | 0.68 (0.57, 0.82)** |
| Higher | 1 | 1 | 1 |
| **Husbands education** | | | |
| No formal | | 0.37 (0.18, 0.65)*** | 0.38 (0.17, 0.61)*** |
| Primary | | 0.47 (0.30, 0.76)*** | 0.44 (0.2, 0.69)*** |
| Secondary | | 0.76 (0.53, 0.97)* | 0.74 (0.50, 0.91)* |
| Higher | | 1 | 1 |
| **Health insurance coverage** | | | |
| No | | 0.62 (0.39, 0.80)* | 0.62 (0.31, 0.76)* |
| Yes | | 1 | 1 |
| **Age** | | | |
| 15–19 | | 0.29 (0.13, 0.49)*** | 0.32 (0.12, 0.53)** |
| 20–24 | | 0.25 (0.13, 0.32)*** | 0.27 (0.11, 0.37)*** |
| 25–29 | | 0.38 (0.22, 0.47)*** | 0.39 (0.20, 0.53)*** |
| 30–34 | | 0.53 (0.35, 0.70)** | 0.54 (0.29, 0.73)** |
| 35–39 | | 0.92 (0.62, 1.22) | 0.93 (0.62, 1.42) |
| 40–49 | | 1 | 1 |
| **Religion** | | | |
| Christian | | 0.93 (0.54, 5.63) | 0.93(0.36, 4.04) |
| Muslim | | 0.55 (0.38, 3.93) | 0.61(0.28, 3.21) |
| Others | | 1 | 1 |
| **Relationship to household head** | | | |
| Head | | 0.80 (0.56, 1.30) | 0.8 (0.49, 1.37) |
| Wife | | 0.84 (0.68, 1.28) | 0.88 (0.73, 1.58) |
| Others | | 1 | 1 |
| **Marital status** | | | |
| Never married | | - | |
| Currently married | | 0.95 (0.56, 1.79) | 0.96 (0.378, 1.479) |
| Previously married | | 1 | 1(,) |
| **Parity** | | | |
| 1 | | 4.29 (3.66, 7.64)*** | 4.10 (3.471, 9.212)*** |
| 2 | | 1.97 (1.78, 3.60)*** | 1.90 (1.752, 4.722)** |
| 3 | | 2.00 (1.48, 2.94) *** | 1.96 (1.56, 4.082)*** |
| 4 | | 1.56 (0.98, 2.01)* | 1.53 (0.989, 2.584)* |
| 5+ | | 1 | 1 |
| **Literacy** | | | |
| No | | 0.98 (0.53, 1.28) | 0.95 (0.504, 1.606) |

(*Continued*)

**Table 4.** (Continued)

| Factors | | Delivery by caesarean section | |
|---|---|---|---|
| | Model 1 | Model 2 | Model 3 |
| Yes | | 1 | 1 |
| **Presence of co-wife** | | | |
| No | | 0.91 (0.72, 1.24) | 0.88 (0.819, 1.704) |
| Yes | | 1 | 1 |
| **Place of residence** | | | |
| Urban | | | 1.40(0.858, 1.589)* |
| Rural | | | 1 |
| **Region** | | | |
| North-central | | | 0.94 (0.547, 1.15) |
| North-east | | | 0.92 (0.583, 1.547) |
| North-west | | | 0.59 (0.387, 1.106)* |
| South-east | | | 1.02 (0.856, 1.903) |
| South-south | | | 1.14 (0.722, 1.538) |
| South-west | | | 1 |

***$P$-value less than 0.001

*$P$-value less than 0.05

Nigeria is among the signatories to the SDGs, which aims to ensure healthy lives and promote wellbeing for all at all ages and leaving no one behind [52, 53]. Specifically, the health target 3.1 aims to achieve a maternal mortality ratio of fewer than 70 deaths per 100,000 live births [1, 52, 53]. Nigeria currently records 814 deaths per 100,000 live births [1]. Getting to 70 deaths per 100,000 live births remains a daunting task for the country. However, with good leadership and effective utilisation of resources, and by prioritising the socially deprived, a well-articulated and implemented health insurance scheme, much progress will be recorded towards achieving the SDG 2030 target. Making maternal health care services, especially birth by caesarean section, universally accessible is paramount to achieving the SDGs target 3.1. This will require not only the effort of the Nigerian government but a global response to address the dire plight of women. This also means bringing the intervention closer to women, that is, providing health care facilities with the capacity to perform caesarean sections in underserved communities. To improve maternal outcomes in Nigeria requires addressing both demand and supply factors hindering access to caesarean births.

### Study strengths and limitations

This study has some limitations. The data were drawn from the NDHS, which was conducted through reliance on women's power of retrospection. This means that women have to recall past events which may have happened over five years. While it is feasible to recall such events, it is also worth noting that it is subject to recall and social desirability bias. The use of caesarean section as an indicator of health inequality is limited because it has been documented that doctors with insufficient experience are left unsupported to make crucial obstetric decisions, resulting in some women being subjected to present and future risks of surgery to be delivered by a caesarean that was not even necessary [51]. This could lead to a higher rate of caesarean section among women who can afford to pay. However, the study's strength lies in the use of a large pool of data, which is rich and nationally representative, to assess inequality in access to

caesarean section as an emergency obstetric service. The use of wealth and education as a proxy measure of inequality is highly reliable.

## Conclusion

This study shows that women in Nigeria have inequitable access to caesarean section service and maternal care. The utilisation of caesarean section among women in the richest households is within the WHO's recommended ideal level, but women in the poorest households fall significantly lower than the recommended rate, and this has catastrophic implications on maternal and neonatal survival. Equity in maternal healthcare is still a promise which has not become a reality; thus, the government need to do more to ensure that maternal health care services are available to those who need them most not only to those who can afford them.

## Acknowledgments

The authors will like to acknowledge the African Population and Health Research Centre for providing the enabling environment for the completion of this work.

## Author Contributions

**Conceptualization:** Boniface Ayanbekongshie Ushie, Ekerette Emmanuel Udoh, Anthony Idowu Ajayi.

**Data curation:** Ekerette Emmanuel Udoh, Anthony Idowu Ajayi.

**Formal analysis:** Ekerette Emmanuel Udoh, Anthony Idowu Ajayi.

**Methodology:** Ekerette Emmanuel Udoh, Anthony Idowu Ajayi.

**Resources:** Anthony Idowu Ajayi.

**Supervision:** Boniface Ayanbekongshie Ushie, Anthony Idowu Ajayi.

**Validation:** Boniface Ayanbekongshie Ushie.

**Writing – original draft:** Boniface Ayanbekongshie Ushie, Ekerette Emmanuel Udoh, Anthony Idowu Ajayi.

**Writing – review & editing:** Boniface Ayanbekongshie Ushie, Ekerette Emmanuel Udoh, Anthony Idowu Ajayi.

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
