## [Editor Report · Decision Letter 0]

13 Jun 2019

PONE-D-19-15544

Examining inequalities in access to delivery by caesarean section in Nigeria

PLOS ONE

Dear Dr Ajayi,

Thank you for submitting your manuscript to PLOS ONE. After careful consideration, we feel that it has merit but does not fully meet PLOS ONE’s publication criteria as it currently stands. Therefore, we invite you to submit a revised version of the manuscript that addresses the points raised during the review process.

We would appreciate receiving your revised manuscript by Jul 28 2019 11:59PM. To enhance the reproducibility of your results, we recommend that if applicable you deposit your laboratory protocols in protocols.io, where a protocol can be assigned its own identifier (DOI) such that it can be cited independently in the future. For instructions see: http://journals.plos.org/plosone/s/submission-guidelines#loc-laboratory-protocols

We look forward to receiving your revised manuscript.

Kind regards,

Charles A. Ameh, PhD, MPH, FWACS (OBGYN), FRCOG

Academic Editor

PLOS ONE

Journal Requirements:

Additional Editor Comments

Dear Authors,

I think this is a very relevant topic, inequity of in the availability of CS and in a country that makes a significant contribution to global maternal mortality. Given that the 2018 Nigeria DHS report and data sets are available, it makes your paper based on data related to 5 years preceding 2013 NDHS less relevant. What about running a similar analysis using DHS 2018 data set and comparing both results? If you can do this, I will be happy to send your revised manuscript for review.

Best wishes
---

## [Author Response · Author response to Decision Letter 0]

15 Jun 2019

Comment

I think this is a very relevant topic, inequity of in the availability of CS and in a country that makes a significant contribution to global maternal mortality. Given that the 2018 Nigeria DHS report and data sets are available, it makes your paper based on data related to 5 years preceding 2013 NDHS less relevant. What about running a similar analysis using DHS 2018 data set and comparing both results? If you can do this, I will be happy to send your revised manuscript for review.

Response

The comments are important and we made effort to use the latest nationally representative data available. We have now contacted DHS and were informed that the NDHS 2018 data is not available to the public at the moment and were also told that only the preliminary result is publicly available. The full report is not yet out. Given the paucity of studies on this important topic, we believe that our paper will contribute to the literature and serve as reference point to future studies. We have however noted the comments and added it as one of our future papers to be written once the NDHS 2018 dataset is finally released.

---

## [Decision Letter · Decision Letter 1]

2 Jul 2019

PONE-D-19-15544R1

Examining inequalities in access to delivery by caesarean section in Nigeria

PLOS ONE

Dear Dr Ajayi,

Thank you for submitting your manuscript to PLOS ONE. After careful consideration, we feel that it has merit but does not fully meet PLOS ONE’s publication criteria as it currently stands. Therefore, we invite you to submit a revised version of the manuscript that addresses the points raised during the review process.

We would appreciate receiving your revised manuscript by Aug 16 2019 11:59PM. To enhance the reproducibility of your results, we recommend that if applicable you deposit your laboratory protocols in protocols.io, where a protocol can be assigned its own identifier (DOI) such that it can be cited independently in the future. For instructions see: http://journals.plos.org/plosone/s/submission-guidelines#loc-laboratory-protocols

We look forward to receiving your revised manuscript.

Kind regards,

Charles A. Ameh, PhD, MPH, FWACS (OBGYN), FRCOG

Academic Editor

PLOS ONE

Additional Editor Comments (if provided):

Thanks for submitting to PLOS One. After careful review, we have concluded that the article does not contribute new knowledge unless significantly revised. A recent article using the same DHS 2013 data set https://bmjopen.bmj.com/content/bmjopen/9/6/e027273.full.pdf may be of interest as you repackage your manuscript.

Reviewers' comments:

Reviewer's Responses to Questions

**Comments to the Author**

1. If the authors have adequately addressed your comments raised in a previous round of review and you feel that this manuscript is now acceptable for publication, you may indicate that here to bypass the “Comments to the Author” section, enter your conflict of interest statement in the “Confidential to Editor” section, and submit your "Accept" recommendation.

Reviewer #1: All comments have been addressed

Reviewer #2: (No Response)

2. Is the manuscript technically sound, and do the data support the conclusions?

Reviewer #1: Yes

Reviewer #2: Yes

3. Has the statistical analysis been performed appropriately and rigorously? 

Reviewer #1: I Don't Know

Reviewer #2: Yes

4. Have the authors made all data underlying the findings in their manuscript fully available?

Reviewer #1: No

Reviewer #2: Yes

5. Is the manuscript presented in an intelligible fashion and written in standard English?

Reviewer #1: No

Reviewer #2: Yes

6. Review Comments to the Author

Reviewer #1: i commend the authors for reviewing an important topic and their ambitious plans to effect a positive change ( improvement in women healthcare in Nigeria) from their findings.

The large data set and parameters reviewed gives strength to this research work. However, there is new and important reference survey from Nigeria DHS 2018 report which makes the research work outdated as it was based on the old NDHS 2013 report. I appreciate the fact that authors have made an attempt to request the data from the 2018 report which is not fully available to the general public. Perhaps this research can be delayed and new analysis done based on the new report

Based on this submission, he conclusion does not add anything new to what is already known about this topic. Linking the disparity to maternity outcome variables within the data set might actually provide strong evidence and attract interest to effect change as intended.

Reviewer #2: There are numerous grammatical errors throughout the text, albeit they do not detract from the meaning. However, they should be addressed. The main issue however is that of the depth of the discussion. That there is inequity across the continuum of care is well addressed in Countdown 2030, and Nigeria remains a Countdown country. Caesarean section rates may be seen to some extent as a surrogate for the accessibility and availability of obstetric care, but there are other important factors to consider, both on the supply and demand sides, that have relevance to the rate of caesarean sections in real life situations. Whereas the authors have focused briefly on some demand side issues that may contribute to women declining a caesarean delivery there are many supply side issues too. I would have been interested to see more of a discussion as to why better educated women in the top wealth quintile have a higher rate of caesarean deliveries as compared to those with the same wealth but less education. Also the issue of whether the woman has health insurance also plays a part. A considered discussion of the influence of these factors on possible decision making processes by doctors is of relevance. Another complicating factor is that not all caesareans are done for good reasons even when the underlying rate is low. There are many reasons for this, but one factor is that doctors with insufficient experience are left unsupported to make crucial obstetric decisions, resulting in some women being subjected to all the present and future risks of surgery to be delivered by a caesarean that wasn't even necessary. This is a limitation of choosing caesarean rates as a surrogate for quality care provision, and should be mentioned. I feel therefore that if caesarean section rate is the measure to be considered in this important discourse on equity than the discussion would greatly benefit from a re-write to look in greater depth at the possible "why" questions that lie behind the inequities. We already know from looking at both Countdown and the DHS that inequity is a big issue, but it is not likely that meaningful change will occur until the "why" questions are addressed more forensically.

7. PLOS authors have the option to publish the peer review history of their article (what does this mean?). If published, this will include your full peer review and any attached files.

Reviewer #1: No

Reviewer #2: No

---

## [Author Response · Author response to Decision Letter 1]

11 Jul 2019

Dear Editor

We are to submit the revised version of our manuscript. The comments provided during the review process have further enabled to strengthen our manuscript. We trust you will find that our manuscript has improved significantly during this process. Our responses to the comments raised by the reviewers are found below.

Best regards 

Anthony Ajayi

On behalf of all authors. 

Reviewer #1: I commend the authors for reviewing an important topic and their ambitious plans to effect a positive change (improvement in women healthcare in Nigeria) from their findings.

The large data set and parameters reviewed gives strength to this research work. However, there is new and important reference survey from Nigeria DHS 2018 report which makes the research work outdated as it was based on the old NDHS 2013 report.

Response 

We thank the reviewer for the constructive and positive feedback.

 I appreciate the fact that authors have made an attempt to request the data from the 2018 report which is not fully available to the general public. Perhaps this research can be delayed and new analysis done based on the new report. Based on this submission, the conclusion does not add anything new to what is already known about this topic. Linking the disparity to maternity outcome variables within the data set might actually provide strong evidence and attract interest to effect change as intended.

Response: We thank the reviewer for the insightful comments. As we said previously, we intend to write a second paper examining the change in inequality in access to birth by caesarean section when the recently completed 2018 DHS data is released and becomes publicly available. We believe our paper fills an important gap given our focus on this important issue that affects the lives of many women in Nigeria as over 58, 000 maternal deaths are recorded annually. Our paper further contributes to the evidence base for understanding the reason for the high burden of maternal deaths in Nigeria. We outline inequality in access to caesarean section as a contributing factor to adverse maternal outcomes in Nigeria, thus, contributing to knowledge on why maternal deaths are prevalent in Nigeria. We hope to build on this study in the future utilizing datasets from previous as well as the 2018 DHS to perform trend analysis to show changes in caesarean section utilization as an emergency obstetric care and hopefully will lead to the desired changes in supply and demand aspect of maternal health delivery in Nigeria. 

Reviewer #2: There are numerous grammatical errors throughout the text, albeit they do not detract from the meaning. However, they should be addressed. 

Response: We have extensively proofread our manuscript and also engaged a language editor. 

Reviewer #2: The main issue, however, is that of the depth of the discussion. That there is inequity across the continuum of care is well addressed in Countdown 2030, and Nigeria remains a Countdown country. Caesarean section rates may be seen to some extent as a surrogate for the accessibility and availability of obstetric care, but there are other important factors to consider, both on the supply and demand sides, that have relevance to the rate of caesarean sections in real life situations. Whereas the authors have focused briefly on some demand side issues that may contribute to women declining a caesarean delivery there are many supply side issues too. I would have been interested to see more of a discussion as to why better educated women in the top wealth quintile have a higher rate of caesarean deliveries as compared to those with the same wealth but less education. Also the issue of whether the woman has health insurance also plays a part. A considered discussion of the influence of these factors on possible decision making processes by doctors is of relevance. Another complicating factor is that not all caesareans are done for good reasons even when the underlying rate is low. There are many reasons for this, but one factor is that doctors with insufficient experience are left unsupported to make crucial obstetric decisions, resulting in some women being subjected to all the present and future risks of surgery to be delivered by a caesarean that wasn't even necessary. This is a limitation of choosing caesarean rates as a surrogate for quality care provision, and should be mentioned. I feel therefore that if caesarean section rate is the measure to be considered in this important discourse on equity than the discussion would greatly benefit from a re-write to look in greater depth at the possible "why" questions that lie behind the inequities. We already know from looking at both Countdown and the DHS that inequity is a big issue, but it is not likely that meaningful change will occur until the "why" questions are addressed more forensically.

Response: We agree with the points highlighted and thank the reviewer for these insightful comments, which have enabled us to significantly improve on the discussion section of our paper. We have revised our discussion to better focus on both supply and demand sides of the caesarean section. Also, we have discussed the effect of health insurance coverage in birth by caesarean section. The countdown 2030 is an enormous piece of work that monitors and measure women’s, children’s, and adolescents’ health in the 81 countries that account for 95% of maternal and 90% of all child deaths worldwide. We have read several publications from the countdown work to gain more insight into the issue of inequity in health outcomes. We have benefited from the insight and experience of the reviewer.

---

## [Decision Letter · Decision Letter 2]

12 Aug 2019

PONE-D-19-15544R2

Examining inequalities in access to delivery by caesarean section in Nigeria

PLOS ONE

Dear Dr Ajayi,

Thank you for submitting your manuscript to PLOS ONE. After careful consideration, we feel that it has merit but does not fully meet PLOS ONE’s publication criteria as it currently stands. Therefore, we invite you to submit a revised version of the manuscript that addresses the points raised during the review process.

We would appreciate receiving your revised manuscript by Sep 26 2019 11:59PM. To enhance the reproducibility of your results, we recommend that if applicable you deposit your laboratory protocols in protocols.io, where a protocol can be assigned its own identifier (DOI) such that it can be cited independently in the future. For instructions see: http://journals.plos.org/plosone/s/submission-guidelines#loc-laboratory-protocols

We look forward to receiving your revised manuscript.

Kind regards,

Charles A. Ameh, PhD, MPH, FWACS (OBGYN), FRCOG

Academic Editor

PLOS ONE

Additional Editor Comments (if provided):

Thanks for making significant improvements to this manuscript. There are minor comments to address before it can be accepted for publication. Thanks

Reviewers' comments:

Reviewer's Responses to Questions

**Comments to the Author**

1. If the authors have adequately addressed your comments raised in a previous round of review and you feel that this manuscript is now acceptable for publication, you may indicate that here to bypass the “Comments to the Author” section, enter your conflict of interest statement in the “Confidential to Editor” section, and submit your "Accept" recommendation.

Reviewer #1: (No Response)

Reviewer #2: (No Response)

2. Is the manuscript technically sound, and do the data support the conclusions?

Reviewer #1: Yes

Reviewer #2: Yes

3. Has the statistical analysis been performed appropriately and rigorously? 

Reviewer #1: Yes

Reviewer #2: Yes

4. Have the authors made all data underlying the findings in their manuscript fully available?

Reviewer #1: Yes

Reviewer #2: Yes

5. Is the manuscript presented in an intelligible fashion and written in standard English?

Reviewer #1: Yes

Reviewer #2: Yes

6. Review Comments to the Author

Reviewer #1: Many thanks to the authors for the revisions made. However, this topic which is based on the old data set (Nigeria DHS 2013) has been covered extensively in some recently published articles therefore adds no new information to what is already known on this topic.

Reviewer #2: This paper has been greatly improved. My only additional comment is that I feel the paragraph in the discussion starting "anecdotal evidence suggests..." where the authors go on to discuss longer hospital stays following caesarean section should have some reference for the assertions made regarding hospital stay. Otherwise, the discussion is much richer and of wider interest as a result.

7. PLOS authors have the option to publish the peer review history of their article (what does this mean?). If published, this will include your full peer review and any attached files.

Reviewer #1: No

Reviewer #2: No

---

## [Author Response · Author response to Decision Letter 2]

13 Aug 2019

Dear Editor,

We want to thank you and the reviewers for the constructive criticism and suggestions that have helped us to improve our manuscript. We have now addressed the comments raised, and we trust you will find our manuscript acceptable for publication. 

Below we provide a point-by-point response to the comments.

Best Regards

Anthony

On behalf of all authors. 

Additional Editor Comments (if provided): Thanks for making significant improvements to this manuscript. There are minor comments to address before it can be accepted for publication. Thanks

Response: We thank the Editor for providing us with constructive feedback that has helped us to improve our manuscript.

Reviewer #1: Many thanks to the authors for the revisions made. However, this topic which is based on the old data set (Nigeria DHS 2013) has been covered extensively in some recently published articles, therefore, adds no new information to what is already known on this topic.

Response: We believe our paper addresses an important topic, and our work is unique by focusing on inequality in access to caesarean section in Nigeria, a country that contributes 19% of global maternal deaths. While the 2013 NDHS may appear to the old in the context of the 2018 NDHS data, the fact is that the 2013 data is still the most recent publicly available. As indicated in our earlier response, when the 2018 data set becomes available, we hope to build on this work to undertake trend analysis on the caesarean section utilization. 

Reviewer #2: This paper has been greatly improved. My only additional comment is that I feel the paragraph in the discussion starting "anecdotal evidence suggests..." where the authors go on to discuss longer hospital stays following caesarean section should have some reference for the assertions made regarding hospital stay. Otherwise, the discussion is much richer and of wider interest as a result.

Response: We appreciate the reviewer’s comments that have enabled us to revise our discussion section significantly. We have now referenced the paragraph as suggested.

---

## [Editor Report · Decision Letter 3]

15 Aug 2019

Examining inequalities in access to delivery by caesarean section in Nigeria

PONE-D-19-15544R3

Dear Dr. Ajayi,

We are pleased to inform you that your manuscript has been judged scientifically suitable for publication and will be formally accepted for publication once it complies with all outstanding technical requirements.

With kind regards,

Charles A. Ameh, PhD, MPH, FWACS (OBGYN), FRCOG

Academic Editor

PLOS ONE

Additional Editor Comments (optional):

Thanks for addressing all comments, I am pleased to recommend this manuscript for publication.
---

## [Editor Report · Acceptance letter]

22 Aug 2019

PONE-D-19-15544R3 

Examining inequalities in access to delivery by caesarean section in Nigeria 

Dear Dr. Ajayi:

I am pleased to inform you that your manuscript has been deemed suitable for publication in PLOS ONE. Congratulations! Your manuscript is now with our production department. 

With kind regards,

on behalf of

Dr. Charles A. Ameh 

Academic Editor

PLOS ONE